# Genomic Variant in NK-Lysin Gene Is Associated with T Lymphocyte Subpopulations in Pigs

**DOI:** 10.3390/genes13111985

**Published:** 2022-10-31

**Authors:** Shifeng Tong, Ningkun Shi, Kaichen Zheng, Zongjun Yin, Xiaodong Zhang, Yang Liu

**Affiliations:** 1Department of Animal Genetics and Breeding, College of Animal Science and Technology, National Experimental Teaching Demonstration Center of Animal Science, Nanjing Agricultural University, Nanjing 210095, China; 2College of Animal Science and Technology, Anhui Agricultural University, Hefei 230036, China

**Keywords:** pig, NK-lysin, SNP, T lymphocyte subpopulations

## Abstract

As an antimicrobial peptide, NK-lysin (*NKL*) plays an important role in the innate immune system of organisms. In this study, 300 piglets (68 Landrace pigs, 158 Large White pigs and 74 Songliao Black pigs) were used to further explore the function of *NLK* gene in porcine immune system. The quantitative real-time PCR analysis detected the *NKL* gene’s expression, and the result demonstrated that *NKL* mRNA was expressed in lung, spleen, stomach, kidney, liver and heart, and the expression level decreased sequentially. A single-nucleotide polymorphism (SNP, g.59070355 G > A) in intron 3 of the *NKL* gene was detected by PCR amplification and sequencing. The results of the Chi-square (χ2) test showed that the genotype of the SNP was consistent with the Hardy-Weinberg equilibrium. What’s more, association analysis results showed the SNP in *NKL* gene was significantly associated with T lymphocyte subpopulations. Different genotypes had significant effects on the proportion of CD4^−^CD8^−^, CD4^−^CD8^+^, CD4^+^CD8^+^, CD8^+^, CD4^+^/CD8^+^ in peripheral blood (*p* < 0.05). These results further suggested that *NKL* could be recognized as a promising immune gene for swine disease resistance breeding.

## 1. Introduction

Antimicrobial peptides (AMPs), a class of peptides closely related to the innate immune system, are widely found in plants and animals to prevent host cells from pathogens through a variety of immune pathways and have been particularly extensively studied in the innate immunity of vertebrates [1]. NK-lysin (*NKL*), first purified in porcine small intestine in 1995, is a 78-residue antimicrobial peptide, and also is an effector molecule by cytotoxic T lymphocytes (CTLs) and natural killer (NK) cells. In addition to the small intestine, *NKL* is also detected in the spleen, bone marrow and colon of pigs [2]. In subsequent studies, *NKL* was found in several species (human [3], bovine [4], chicken [5], quail [6], et al.) and has been shown to play an important role in animal immune system. *NKL* effectively inhibits the activity of bacteria (Gram-positive and Gram-negative bacteria), fungi, viruses and pathogens [7,8]. What’s more, it has antitumor activity and can kill many types of cancer cells [9,10]. In recent studies, *NKL* has also been shown to have immunomodulatory effects [11].

Several studies have shown that genetic polymorphisms in *NKL* affect the immune function of animal organisms, especially in vertebrates. A single-nucleotide polymorphism (SNP) on the *NKL* open reading frame (ORF) was identified in different chicken species, which resulted in an amino acid change from Asn (N) to Asp (D), and the nucleotide mutation reduced antibacterial and anticancer activities [10]. Similar to what was observed in chickens, 9 SNPs were found in the ORF of Japanese quail *NKL*, and these SNPs also affected the antimicrobial properties of the antimicrobial peptides [12]. In most mammals, *NKL* is encoded by a single gene, but four *NKL* genes (*NK1*, *NK2A*, *NK2B* and *NK2C*) were identified on chromosome 11 in cattle, *NK2A*, *NK2B* and *NK2C* are generated by *NKL* gene copy number variation. And all 4 genes are transcribed, and their encoded products are repressive against both Gram-positive and Gram-negative bacteria [8]. These findings suggested that *NKL* is a vital candidate gene which is positional and functional for immune traits. Even though *NKL* has been identified in pigs, the studies of genetic polymorphisms and its immune response remain inadequate.

To figure out a deeper understanding of porcine *NKL* gene involved in innate immune system, we analyzed the expression patterns and polymorphisms of porcine *NKL* gene. T lymphocyte subpopulations, as important effectors of the animal’s innate immune system, perform different functions in the immune system. Therefore, we further explored the associations between the SNP of *NLK* gene and T lymphocyte subpopulations of the 3 pig populations.

## 2. Materials and Methods

### 2.1. Animal Population and Sample Collection

The animal population in this study consisted of 300 piglets, including 68 Landrace piglets, 158 Large White piglets and 74 Songliao Black piglets. All piglets were reared uniformly under the same standard conditions in the experimental farm of the Institute of Animal Science, Chinese Academy of Agricultural Sciences, Beijing, China. All piglets were vaccinated with Classical Swine Fever (CSF) live vaccine at 21 days of age to produce high titers of neutralizing antibodies, which could be used as a “stress” index to detect the immune response of lymphocyte subpopulations [13,14].

The ear tissue samples of each individual were collected in 75% alcohol and stored in a refrigerator at −20 °C for DNA extraction. The blood samples of all piglets were collected one day before vaccination (20 days old) and two weeks after vaccination (35 days old) for the determination of T lymphocyte subpopulation levels. In addition, three 35-day-old Landrace piglets were selected for the slaughtering experiment, and 7 kinds of tissues (heart, liver, spleen, stomach, lung, kidney, muscle) were collected within 30 min after slaughter, then stored in −80 °C refrigerator after liquid nitrogen freezing for RNA extraction.

### 2.2. Measurement of T Lymphocyte Subpopulation Levels in Peripheral Blood

The T lymphocyte subpopulation levels were detected in Beijing Xiyuan Hospital of Chinese Academy of Medical Sciences, and the specific detection information can be found in our previous study [15]. The T lymphocyte subpopulation levels were detected by EPICS Flow Cytometer with an argon laser with an excitation wavelength of 488 nm and using FITC-CD4/PE-CD8 monoclonal antibody kits. In this study, the levels of seven kinds of T lymphocyte subpopulation: proportions of CD4^−^CD8^−^, CD4^+^CD8^+^, CD4^+^CD8^−^, CD4^−^CD8^+^, CD4^+^, CD8^+^, and the ratio of CD4^+^ to CD8^+^ (CD4^+^/CD8^+^).

### 2.3. DNA Extraction and cDNA Preparation

Genomic DNA of piglet ear tissue was extracted by phenol-chloroform method. Total RNA was extracted from various tissues with Trizol reagent (Invitgen, Carlsbad, CA, USA). The quality of DNA and RNA was verified by 1% agarose gel electrophoresis. The concentration of DNA and RNA was measured by NanoDrop 2000 spectrophotometer (Thermo Scientific, Waltham, MA, USA). According to the manufacturer’s instructions, RNA was purified and reversely transcribed into cDNA by using PrimerScript^®^ RT reagent Kit with gDNA Eraser (Takara Biotechnology, Dalian, China) and stored in a refrigerator at −20 °C.

### 2.4. Quantitative Real-Time PCR Analysis

Quantitative real-time PCR analysis (qPCR) was used to determine the total mRNA expression level of *NKL* gene in 7 various tissues of piglets by LightCycler^®^ 480 II instrument (Roche Diagnostics GmbH, Penzberg, Germany). Here is the qPCR reaction system: 20 ng of cDNA, 10 μL of 2 × SYBR green I mixture, 10 pM of the forward and reverse primers respectively, and add RNA-free water to final volume is 20 μL. The primers of *NKL* (F: 5′-AACCCCAGGCTATCTGTGTG-3′, R: 5′-AGGGTGCTGGAGTTTCTGTG-3′.) and *GAPDH* (F: 5′- GTCCACTGGTGTCTTCACGA-3′, R: 5′-GCTGACGATCTTGAGGGAGT-3′.) were synthesized by Sangon Biotech (Shanghai, China). PCR amplification is carried out according to the following reaction condition: 95 °C for 10 min, 45 cycles of 95 °C for 10s, 60 °C for 10 s and 72 °C for 10 s. All measurements were carried out in triplicate and normalized to GAPDH by the 2^−^^△△Ct^ method.

### 2.5. SNP Identification and Genotyping

Log in to GenBank and download the porcine *NKL* genomic sequence (GenBank Accession Number: NC_010445.4). The primers of all exons and partial adjacent introns of *NKL* gene were designed by Primers 5.0 software (Table 1). The primers were synthesized by Sangon Biotech (Shanghai, China).

DNA samples from 30 piglets were randomly selected to establish an equal amount of DNA mixed pool with 50 ng/L DNA concentration in each individual. The sample of DNA mixed pool was used as a template for PCR reaction. Here is the PCR reaction system (25 μL): 0.125 μL of TaKaRa Taq HS (5 U/μL; Takara Biotechnology, Dalian, China), 5 μL of 10 × PCR Buffer (Mg^2+^ plus), 2 μL of dNTP Mixture (2.5 mM), 50 ng of DNA samples, 10 pM of forward and reverse primers, ddH2O up to 25 μL. The following reaction conditions need to be followed correctly: denaturation at 94 °C for 5 min, then by 34 cycles at 94 °C for 30 s, 59–60.1 °C for 35 s, 72 °C for 35 s, and finally extend at 72 °C for 5 min. The qualified PCR products were sequenced by Sangon Biotech (Shanghai, China), and the sequencing results were analyzed, and SNPs detection were conducted by Chromes 2.3.1 and DNAMAN 6.0 software. According to the results of SNP identification, the SNPs from 300 piglets were genotyped by MALDI-TOF MS (Sequenom MassARRAY^®^, Bioyong Technologies Inc., Beijing, China).

### 2.6. Association Analysis

SAS 9.2 software was used to analyze the association between the genotypes of SNP and the immune traits of T lymphocyte subpopulations. The models used were as follows:y=μ+Xβ+Tv+bm+Za+e
where *y* is the vector of phenotypic values of innate immune traits analyzed; μ is the overall mean; *β* is the vector of fixed effects including sex, sampling batch; *v* is the vector of random litter effects; *m* is the vector of SNP genotype with three levels; *a* is the vector of the residual polygenetic effects with $a\sim N(0,A\sigma_{a}^{2})$, where *A* is the numerator relationship matrix; *X*, *T* and *Z* are the incidence matrices for *β*, *v* and *a*, respectively; b is the regression coefficient of phenotypes on SNP genotypes; *e* is the vector of residual errors with $e\sim N\left(0,I\sigma_{a}^{2}\right)$.

## 3. Results

### 3.1. The Relative Expression Level of NKL Gene in 7 Various Tissues

The results of qPCR showed that total mRNA of the *NKL* gene was expressed in all six tissues analyzed, but not in skeletal muscle. What’s more, the *NKL* mRNA was expressed at the highest level in the lung, which was significantly higher than in other tissues. The relative expression levels of *NKL* mRNA were sequentially lower in the spleen, stomach, kidney and liver, and the lowest in the heart (Figure 1).

### 3.2. Polymorphism Detection and Association Analysis of Porcine NKL Gene

Porcine *NKL* gene (GenBank Accession Number: NC_010445.4) is located on chromosome 7 and consists of 5 exons and 4 introns. One SNP (g.59070355 G > A) in intron 3 was detected through direct sequencing of PCR products. Genotyping results of 300 piglets at the SNP showed that 3 genotypes (GG, GA and AA) of this SNP were present in Landrace pigs, Large White pigs and Songliao Black pigs. And the allele G was the dominant allele; on the contrary, the frequency of allele A was lower, especially the frequency of GG genotype was the highest and the frequency of AA genotype was lower detected in all three pig populations. What’s more, the results of the Chi-square (χ^2^) test showed that the genotype of the SNP was consistent with the Hardy-Weinberg equilibrium (Table 2).

Association analysis was performed between the traits of 7 T-lymphocyte subpopulations from 300 piglets of three pig breeds and the SNP (g.59070355 G > A) for the *NKL* gene (Table 3). The results revealed that the proportion of CD4^−^CD8^−^, CD4^−^CD8^+^, CD4^+^CD8^+^, and CD8^+^ in peripheral blood of individuals with GG and GA genotypes were significantly higher than those of individuals with AA genotype (*p* < 0.05), but the ratio of CD4^+^ to CD8^+^ in peripheral blood of individuals with GG genotype was significantly lower than that of pigs with AA genotype (*p* < 0.05), and the ratio of CD4^+^ to CD8^+^ was not significantly different between individuals with GA genotypes and AA genotypes. In addition, the traits of other 6 T-lymphocyte subpopulations were not significantly different between individuals with GG genotypes and GA genotypes.

## 4. Discussion

### 4.1. T Lymphocytes Traits

T lymphocytes are important immune cells which play a vital part in cell-mediated immunity. T lymphocytes expressing CD4 membrane markers are defined as T-helper lymphocytes, which are necessary for regulating both the humoral and cellular immune response [16,17].In the contrary, T lymphocytes expressing CD8 are defined as T-cytotoxic lymphocytes, which can process cytolytic proteins to kill pathogen-challenged cells by initiating an apoptotic program [18]. The specific expression of CD4 and CD8 antigens in the porcine immune system defines 4 extra-thymic T lymphocyte subsets, including CD4^−^CD8^−^, CD4^+^CD8^+^, CD4^−^CD8^+^, CD4^+^CD8^−^ T lymphocytes [19,20]. Since the proportion of the 4 T lymphocyte subsets in the peripheral blood changes in response to the immune response, it can be used as an index of immune response capacity. In our studies, the proportion of all 4 T lymphocyte subpopulations in the peripheral blood of pigs with GG and GA genotypes was higher than that of AA genotypes, which may indicate that pigs with GG and GA genotypes have higher immune response capacity than those with AA genotypes. However, the ratio of CD4^+^ to CD8^+^ may affect T lymphocyte immune functions, and a low ratio of CD4^+^ to CD8^+^ has been reported to be associated with high classical swine fever virus (CSFV) replication [21]. Our results showed that the ratio of CD4^+^ to CD8^+^ in brood of pigs with genotype GG was the lowest, which implied that pigs with genotype GG may have a weaker immune response during CSFV infection than those with the other two genotypes.

### 4.2. Association Analysis

Through qPCR analysis, we found that *NKL* mRNA was expressed in six tissues of piglets (lung, spleen, stomach, kidney, liver and heart), its expression level decreased sequentially in the six tissues, and the trend of mRNA expression was consistent with the research of Wang et al. [14]. Previous studies have indicated that mutations can markedly influence mRNA secondary structure [22], resulting in mRNA degradation [23] and alteration of protein translation [24]. In this study, the SNP (g.59070355 G > A) of *NKL* gene was detected and the SNP is a synonymous mutation, which does not cause changes in the encoded amino acids. However, accumulating studies have shown that synonymous mutations are no longer silent mutations. Cheng et al. [25] confirmed that synonymous mutations alter the expression of porcine *IGF1* gene and affect protein folding. Synonymous mutations are also closely related to human diseases, which can influence on the occurrence of diseases by affecting mRNA stability, protein translation and folding [26]. In a large number of biological genomes, it has been found that codon usage bias, the preferential use of some synonymous codons, will affect the structure and expression level of proteins [27]. It has been proved that codon use can change the translation elongation rate of proteins in both Neosporidia and Drosophila [28,29]. Codon usage bias also has an important impact on the global translation efficiency and cellular adaptability of Escherichia coli [30]. Furthermore, codon usage bias can also affect protein structure and expression level by adjusting translation accuracy, and the levels of mRNA and tRNA [27]. Consequently, the synonymous G/A substitution of the *NKL* gene may be a potential functional variant, which may affect porcine immune function by changing gene expression and protein structure.

## 5. Conclusions

In conclusion, *NKL* was widely expressed in all tissues analyzed (lung, spleen, stomach, kidney, liver and heart) except skeletal muscle, and the expression level decreased sequentially. One SNP (g.59070355 G > A) was detected in intron 3 of *NKL* gene, which was significantly correlated with the proportion of CD4^−^CD8^−^, CD4^+^CD8^+^, CD4^−^CD8^+^, CD8^+^, and CD4^+^/CD8^+^ in peripheral blood. Thus, it can be seen that *NKL* gene may be used as a candidate gene to regulate T lymphocyte subpopulation in porcine peripheral blood and be used in marker-assisted selection in pig breeding.

## Figures and Tables

**Figure 1 genes-13-01985-f001:**
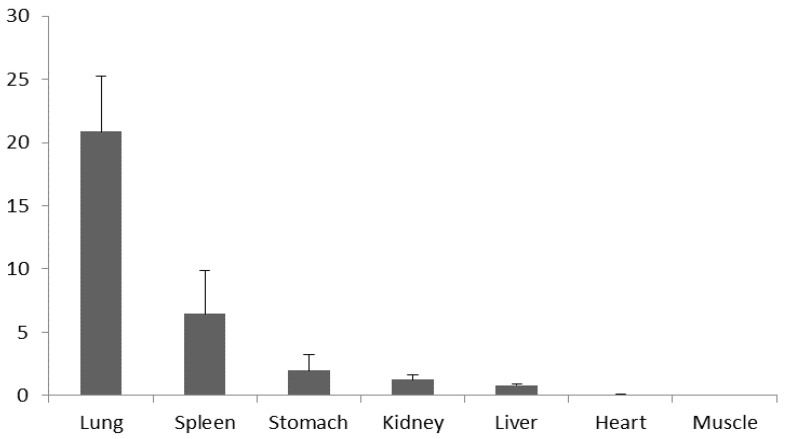
Relative quantification of total mRNA expression levels of porcine NK-lysin (*NKL*) gene in 7 tissues. Bars represent the Mean ± SEM (n = 3).

**Table 1 genes-13-01985-t001:** Primers and annealing temperatures used for PCRs of the porcine NK-lysin (*NKL*) gene.

Primers	Sequences 5′–3′	Anneal Temperature	Product Size
F1	AGACAAGGAGGGCAGAGGAG	59.0 °C	530 bp
R1	CTTCCCACCAGCTGTCTCTC		
F2	CTCCATCTGCTCCATCTGCT	60.0 °C	360 bp
R2	AGAGATCTCCAGCCCTCACC		
F3	CAGCTAGCCTGGTTCAGGTC	60.1 °C	420 bp
R3	TGAAGTCCATCTGCTGACCA		
F4	TCCTTGTCCCCAACACTTTC	59.5 °C	480 bp
R4	CCCACTTTTCAGGTTGCTGT		
F5	ATCCTTCACCCACTGACCAA	60.0 °C	420 bp
R5	AGGGTGCTGGAGTTTCTGTG		

**Table 2 genes-13-01985-t002:** Genotype frequencies and allelic frequencies of *NK-lysin (NKL)* gene in three pig populations.

Breed	Number	Genotype Frequencies			Allele Frequencies		χ^2^ (P)
		GG	GA	AA	G	A	
Landrace	68	0.485 (33)	0.441 (30)	0.074 (5)	0.706	0.294	2.254 (0.324)
Large White	158	0.639 (101)	0.285 (45)	0.076 (12)	0.781	0.219	5.107 (0.078)
Songliao Black	74	0.460 (34)	0.351 (26)	0.189 (14)	0.636	0.364	1.014 (0.602)

**Table 3 genes-13-01985-t003:** Association analysis and multiple tests of the SNP (g.59070355 G > A) of NK-lysin (*NKL*) gene with immune traits in three pig populations.

Traits	Genotypes (Means ± Standard Error of Means)		
	GG (n = 168)	GA (n = 101)	AA (n = 31)
CD4^−^CD8^−^	36.541 ± 7.391 ^a^	35.058 ± 5.386 ^a^	33.365 ± 4.965 ^b^
CD4^+^CD8^−^	14.997 ± 3.365	13.819 ± 3.014	14.007 ± 3.275
CD4^−^CD8^+^	39.917 ± 8.307 ^a^	38.382 ± 7.124 ^a^	36.324 ± 6.813 ^b^
CD4^+^CD8^+^	36.676 ± 6.804 ^a^	35.758 ± 7.563 ^a^	34.542 ± 6.565 ^b^
CD4^+^	26.912 ± 4.971	27.079 ± 4.413	26.740 ± 4.597
CD8^+^	51.117 ± 9.472 ^A^	48.087 ± 8.454 ^A^	46.598 ± 7.541 ^B^
CD4^+^/CD8^+^	0.53 ± 0.042 ^a^	0.56 ± 0.056 ^b^	0.56 ± 0.053 ^b^

^a,b^, different superscripts in the same line represent significant differences (*p* < 0.05); ^A,B^, Different superscripts in the same line represent extremely significant differences (*p* < 0.01); and the same superscript or no marked represents no significant difference.

## Data Availability

The authors declare that all data supporting the findings of this study are available within the article or from the corresponding author upon reasonable request.

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
