# Peer review of "Genomic Variant in NK-Lysin Gene Is Associated with T Lymphocyte Subpopulations in Pigs"

_genes, 2022, doi:10.3390/genes13111985_

Round 1
Reviewer 1 Report
The paper "Genomic variant in NK-lysin gene is associated with T lymphocyte subpopulations in pigs" investigated the gene expression and role of an SNP in NK-lysin (NKL) gene in T cells subpopulations in pigs.
I would like to thank all the authors for submitting this paper. However, in my opinion, these types of studies are a bit old-fashioned.
Here are some comments:
- The language of the paper must be improved. There are plenty of grammar errors in the text.
- When talking about immune cells, you have to show the gating strategies (for flow cytometry or Diff. Readers like to see how you define the subpopulations. Also, I don't think looking at double+ and double negatives in the blood can tell you anything about the immune cell functions and roles. The ratio of CD4/CD8 is the best comparison.
Good luck
Author Response
Response to the reviewer’s comments on our manuscript:
We appreciate the valuable comments received from section editor and the two reviewers. They have helped us to significantly improve our study.
Reviewer(s)' Comments to Author:
The paper "Genomic variant in NK-lysin gene is associated with T lymphocyte subpopulations in pigs" investigated the gene expression and role of an SNP in NK-lysin (NKL) gene in T cells subpopulations in pigs.
I would like to thank all the authors for submitting this paper. However, in my opinion, these types of studies are a bit old-fashioned.
Here are some comments:
The language of the paper must be improved. There are plenty of grammar errors in the text.
Respond: Thanks for the reviewer’s suggestion. Corrected carefully as suggested.
When talking about immune cells, you have to show the gating strategies (for flow cytometry or Diff. Readers like to see how you define the subpopulations. Also, I don't think looking at double+ and double negatives in the blood can tell you anything about the immune cell functions and roles. The ratio of CD4/CD8 is the best comparison.
Respond: Thanks for the reviewer’s suggestion and we added the detection introduction part. The T lymphocyte subpopulation levels were detected by EPICS Flow Cytometer with an argon laser with an excitation wavelength of 488 nm and using FITC-CD4/PE-CD8 monoclonal antibody kits. (Materials and Methods 2.2)
In addition, yes, the ratio of CD4/CD8 is the best comparative trait, as shown in our results (Table 3). According to previous studies, we can know that T lymphocyte subsets are more diverse in the immune system of pigs. In addition to CD4-CD8+ and CD4+CD8- T lymphocytes, CD4-CD8- and CD4+CD8+ lymphocyte subpopulations are prominent in blood as well as in porcine lymphoid tissues. Functional analyses of porcine T-lymphocyte subpopulations revealed cells derived from the CD4+CD8+ T-helper-cell subpopulation were reactive in primary immune responses in vitro and able to respond to recall antigen in a secondary immune response.
Please refer to the two papers:
- Functional characterization of porcine CD4+CD8+ extrathymic T lymphocytes. Cell Immunol. 1996, 168(2):291-6. doi: 10.1006/cimm.1996.0078.
- Characterization of porcine T lymphocytes and their immune response against viral antigens. J Biotechnol. 1999, 73(2-3):223-33. doi: 10.1016/s0168-1656(99)00140-6.

Reviewer 2 Report
This paper is considered to be an interesting paper on genomic analysis on innate immune traits of pig. In this study, the purpose of the study is very impressive and I think that important data have been obtained. However, I think that this study used only small data and authors did not show clear conclusion in this study.
1. (Method, Line 127 & Result Table03) What is phenotypic values and meaning of innate immune traits in this model? and What information would author like to know from this model? And I wonder how author made this model.
2. (Result, Line 136) Why is the expression of NKL gene different in each tissue?
3. (Result, Line 147) I think there are various SNPs in the NKL gene. What is the reason for choosing g.59070355 G>A?
4. (Discussion, Line 173) Analysis was performed using a linear model. Does each trait data have a normal distribution? It is necessary to show information about the entire phenotypic data. Additionally, author used several traits, but did authot confirm that the traits had an independent relationship?
5. In this paper, I did not show a clear conclusion on the interpretation of the relationship between the phenotype and the genotype of the interest SNP. As a result of statistical analysis, rather than presenting simple results, it is necessary to suggest what kind of relationship there is with genotype and phenotype.
Author Response
Response to the reviewer’s comments on our manuscript:
We appreciate the valuable comments received from section editor and the two reviewers. They have helped us to significantly improve our study.
Reviewer(s)' Comments to Author:
This paper is considered to be an interesting paper on genomic analysis on innate immune traits of pig. In this study, the purpose of the study is very impressive and I think that important data have been obtained. However, I think that this study used only small data and authors did not show clear conclusion in this study.
- (Method, Line 127 & Result Table03) What is phenotypic values and meaning of innate immune traits in this model? and What information would author like to know from this model? And I wonder how author made this model.
Respond: Thanks for the reviewer’s suggestion. The levels of seven kinds of T lymphocyte subpopulation are phenotypic values in the model. For this model, we can analyze the relationship between different SNP genotypes and the level of T lymphocyte subpopulations, and further reveal the effect of SNP in NKL gene on immune function. (L137, 180)
- (Result, Line 136) Why is the expression of NKL gene different in each tissue?
Respond: Thanks for the reviewer’s suggestion. As we know, the expression level of NKL gene is different in different tissues and different time. Related studies have shown that NKL gene also shows different expression levels in different tissues of bison, Cyprinus carpio and Larimichthys crocea. (Reference: doi: 10.1093/jhered/esy022; doi: 10.1016/j.fsi.2017.11.030; doi: 10.1016/j.fsi.2016.05.035.) (L149)
- (Result, Line 147) I think there are various SNPs in the NKL gene. What is the reason for choosing g.59070355 G>A?
Respond: Thanks for the reviewer’s suggestion. In this study, after scan region of all exons and introns of NKL gene through direct sequencing of PCR products, we found only one SNP in our three experimental pig populations. (L158)
- (Discussion, Line 173) Analysis was performed using a linear model. Does each trait data have a normal distribution? It is necessary to show information about the entire phenotypic data. Additionally, author used several traits, but did authot confirm that the traits had an independent relationship?
Respond: Thanks for the reviewer’s suggestion. According to the suggestion, we collected a little error in the mixed model (Part 2.6). In addition, yes, each trait has a normal distribution and they have an independent relationship, as shown in the Figure 1. The traits can be found in our previous study (Wenwen, Wang, Yang Liu, et al. Single-nucleotide polymorphisms in CD8A and their associations with T lymphocyte subpopulations in pig. Molecular Genetics & Genomics, 2015. DOI: 10.1007/s00438-015-1008-8)
Figure 1 CD4 versus CD8 expression of the un-separated T lymphocyte was detected by Flow Cytometer. The fluorescence intensities of CD8 and CD4 are displayed in two-dimensional contour plot.
- In this paper, I did not show a clear conclusion on the interpretation of the relationship between the phenotype and the genotype of the interest SNP. As a result of statistical analysis, rather than presenting simple results, it is necessary to suggest what kind of relationship there is with genotype and phenotype.
Respond: Thanks for the reviewer’s suggestion. Our clear conclusion is NKL gene may be used as a candidate marker to regulate T lymphocyte subsets in pig breeding. Result 3.2 showed that different genotypes of SNP had significant effects on the level of T lymphocyte subpopulations in peripheral blood of piglets. The reason for the effect of SNP on T lymphocyte subpopulations is expounded in the discussion section. However, the pathway in which the SNP affects the level of T lymphocyte subpopulations remains to be further studied.
Reviewer 3 Report
This article provides new information for the function of NK-lysin in porcine immune system. In this study, the purpose of the study is very interesting and I think that important data have been obtained. However, there are still some shortcomings in this study that must to be improved.
Here are some comments:
1. The language and format of this article need to be improved. There are some grammar and format errors in the text.
2. The gene names should be in italic format.
3. There should be many kinds of SNPs in NKL gene. In this study, only one SNP was found in intron 3, and the effect of SNP in intron on gene function was lower than that in CDS region and gene regulatory region. Has the author found SNP in other regions?
Author Response
Response to Reviewer 3 Comments
We appreciate the valuable comments received from section editor and the two reviewers. They have helped us to significantly improve our study.
Point 1: The language and format of this article need to be improved. There are some grammar and format errors in the text.
Respond 1: Thanks for the reviewer’s suggestion. Corrected carefully as suggested.
Point 2: The gene names should be in italic format.
Respond 2: Thanks for the reviewer’s suggestion. Corrected carefully as suggested.
Point 3: There should be many kinds of SNPs in NKL gene. In this study, only one SNP was found in intron 3, and the effect of SNP in intron on gene function was lower than that in CDS region and gene regulatory region. Has the author found SNP in other regions?
Respond 3: Thanks for the reviewer’s suggestion. In this study, after scan region of all exons and introns of NKL gene through direct sequencing of PCR products, we found only one SNP (g.59070355 G>A) in intron 3 of the NKL gene. SNP in intron can affect gene function by affecting the activity of splice site, transcriptional level and translation level. No SNP was found in other regions, which may be related to the samples used in this study.

Round 2
Reviewer 1 Report
Thank you for providing the update. However, I am still not convinced that the apper has the qualification to be accepted in this reputed journal. The Authors may consider submitting the paper to other jouranls. Good luck.
Author Response
Thanks for the reviewer’s comments.
Reviewer 2 Report
The authors did their best to answer all the questions.
Author Response
Thanks for the reviewer’s comments.